# Mathematical-based morphological classification of skin eruptions corresponding to the pathophysiological state of chronic spontaneous urticaria

Sungrim Seirin-Lee[1,2,3✉], Daiki Matsubara[4], Yuhki Yanase[3,5], Takuma Kunieda[5], Shunsuke Takahagi [3,4✉] & Michihiro Hide [4,6]

## Abstract

**Background** Chronic spontaneous urticaria (CSU) is one of the most intractable human-specific skin diseases. However, as no experimental animal model exists, the mechanism underlying disease pathogenesis in vivo remains unclear, making the establishment of a curative treatment challenging.

**Methods** A novel approach combining mathematical modelling, in vitro experiments and clinical data analysis was used to infer the pathological state of CSU patients from geometric features of the skin eruptions.

**Results** Based on our hierarchical mathematical modelling, the eruptions of CSU were classified into five categories, each with distinct histamine, basophils, mast cells and coagulation factors network signatures. The analysis of 105 real CSU patients with this classification by six individual dermatologists achieved 87.6% agreement. Furthermore, our network analysis revealed that the coagulation status likely determines boundary/area pattern of wheals, while the state of spontaneous histamine release from mast cells may contribute to the divergence of size and outline of the eruptions.

**Conclusions** Our multi-faceted approach was accurate in defining pathophysiological states of disease based on geometric features offering the potential to improve the accuracy of CSU diagnosis and better management of the disease in the clinic.

## Plain language summary

Chronic spontaneous urticaria (CSU) is a persistent skin disease that causes red itchy skin eruptions, called wheals, that are of various shapes. These wheals repeatedly appear and disappear daily for up to weeks or even decades, severely impacting the quality of life of those affected. The causes, consequences or disease processes are largely understudied. Here, we developed a novel approach using mathematical modelling to analyse the geometric measurements of patients' wheals alongside laboratory experiments on human skin samples with eruptions to improve CSU diagnosis and clinical management. We find that geometric measurements of these skin eruptions can be classified into five categories, which could facilitate accurate diagnosis and consequently better clinical management of CSU and may be more widely applied to other skin diseases.

[1] Kyoto University Institute for the Advanced Study of Human Biology (ASHBi), KUIAS, Kyoto University, Yoshida-Konoe-cho, Sakyo-ku, Kyoto 606-8501, Japan. [2] Department of Mathematical Medicine, Graduate School of Medicine, Kyoto University, Yoshida-Konoe-cho, Sakyo-ku, Kyoto 606-8501, Japan. [3] JST CREST, 4-1-8 Honcho, Kawaguchi, Saitama 332-0012, Japan. [4] Department of Dermatology, Institute of Biomedical & Health Sciences, Hiroshima University, 1-2-3 Kasumi, Minami-ku, Hiroshima 734-8551, Japan. [5] Department of Pharmacotherapy, Graduate School of Biomedical Sciences, Hiroshima University, 1-2-3 Kasumi, Minami-ku, Hiroshima 734-8551, Japan. [6] Department of Dermatology, Hiroshima City Hiroshima Citizens Hospital, Motomachi, Naka-ku, Hiroshima 730-8518, Japan. ✉email: lee.seirin.2c@kyoto-u.ac.jp; shunstk@hiroshima-u.ac.jp

The skin is the largest organ in the human body and plays an important role in maintaining homeostasis and protecting the body from the external environment. Antigens, chemicals, and ultraviolet rays, and endogenous substances can cause cutaneous inflammatory reactions and tumorigenesis. A variety of skin diseases have an impact on society due to seriously impairing the quality of life of patients, even if they are not life-threatening. Urticaria is one of the most common human skin disorders, affecting at least one in five people during their lifetime[1,2]. It is characterized by the appearance of skin eruptions, also known as wheals, that are of various shapes ranging from a few millimeters to several centimeters in size and is typically accompanied by itching. Patients with urticaria suffer from these symptoms potentially for years to even decades. Among the different subtypes of urticaria, chronic spontaneous urticaria (CSU) is characterized by the spontaneous appearance and disappearance of wheals[3].

The pathophysiological states associated with CSU include autoimmune responses, cellular infiltrates, coagulation, and the complement system[4]. However, how these responses contribute to wheas remains unknown, which is largely due to the limited understanding and scarcity of clinical data on the in vivo pathological structure of urticaria. Furthermore, due to the human-specific nature of the disease, there is also a lack of an appropriate animal model system that can recapitulate the entire disease pathology[5]. Consequently, elucidating the underlying pathophysiological dynamics in vivo remains challenging, and developing patient-specific effective treatments for urticaria remains difficult.

To overcome the limitations of the lack of human-specific skin disease models, we developed a novel approach linking the geometric features of skin eruption with pathophysiological structures in vivo. As all skin diseases are characterized by "visible outputs", skin eruptions can be considered a reflection of in vivo pathophysiological dynamics. However, eruption geometry has neither been fully understood nor explored as a measure of the differences in pathological states between patients.

Thus, in this study, we focus on the morphological features of skin eruptions and develop a novel approach linking the eruption geometry of patients to the in vivo pathological dynamics of CSU using hierarchical mathematical modeling, in vitro experiments, and clinical data analysis of the eruption patterns of urticaria. We show that the major in vivo pathophysiological state of CSU can be typically defined as one of five skin eruption pattern types, thereby predicting degrees of contribution by each pathogenetic factor in urticaria. This study suggests that mathematical modeling analysis of morphology appearing on the body may help not only with diagnosis and classification but also reveal molecular mechanisms and targets for new treatments of a variety of human diseases that develop geometric lesions.

## Methods

**Collection of images of wheals and their morphological classification.** Images of skin eruptions (wheals) were collected from 105 patients with CSU who visited the Department of Dermatology at Hiroshima University Hospital between January 1, 2000 and April 15, 2022. The institutional ethics review board approved the study protocol (Ethical Committee for Epidemiology of Hiroshima University, Hiroshima, Japan, approval number; E2020-2388), which involved analyzing images stored in the secured hospital database. Informed consent was obtained in the form of opt-out on the website. Six dermatologists, specializing in allergic skin diseases, classified the morphology of wheals on each CSU patient into one of the following seven major categories: annular, broken-annular, geographic, circular, dot, uniform

(shapeless and spread throughout the body), and none of the above (NA).

The uniform and NA cases were considered unclassifiable based on the EGe criteria; therefore, six categories were used in the final analysis. Sample number 105 exceeded the minimal number required for an error with the true classification result within 8% and the significance level ($\alpha$) was 5%[6].

**Statistical analysis.** To classify the eruption patterns of patients, we used the vote-counting method, performed by six dermatologists. The reliability **SB** was calculated as follows:

$$\mathbf{SB} = \frac{\text{Number of doctors who selected the type in EGe criteria}}{\text{Total number of doctors who participated in the classification}}. \tag{1}$$

**SB** is the probability of $p(A \wedge B)/p(A)$ where $p(A)$ and $p(B)$ are the probabilities that eruption patterns are classified in one of two classes or one of five types by doctors, respectively (Table 1). The reliability **SS** was calculated as follows:

$$\mathbf{SS} = p(A) \times \mathbf{SB} = p(A) \times \frac{p(A \wedge B)}{p(A)} = p(A \wedge B). \tag{2}$$

**SS** is the probability that both the class and type of eruption pattern were chosen correctly and that the **SS**-based analysis would provide more reliable statistical results. To confirm the statistical verification of the classifiable case data, we calculated the $P$-value as follows:

$$P_{value} = \sum_{k=k_0}^{n} nC_k p_0^k (1 - p_0)^{n-k}, \tag{3}$$

where $k_0$ is the sample number of classifiable cases, $n$ is the total sample number, and $p_0$ is the probability of the null/alternative hypothesis that the classifiable rate in the population is below/above $p_0$ %.

**Function representing the skin eruption state.** To reflect the eruption state of the skin ($\widetilde{\Omega}$) based on the molecular dynamics of the dermis ($\Omega_D$), we define the eruption state function, $S_w : \Omega_D \rightarrow \widetilde{\Omega}(\subset \mathbb{R}^2)$, as follows:

$$S_w([A]) = \frac{1}{1 + \exp\left[-\beta_w([A] - [A]_r)\right]}. \tag{4}$$

This function connected the dynamics of the coagulation factors ($[A](\mathbf{x}, t) \equiv [C](\mathbf{x}, t)$) leaked from blood vessels or histamine ($[A](\mathbf{x}, t) \equiv [H_M](\mathbf{x}, t)$) released from mast cells in the dermis to the state of wheal formation seen on the skin ($\widetilde{\Omega}$). $\beta_w$ is the positive value and $[A]_r$ is the threshold value at which the eruption appears on the skin. This function allows for two states of eruption states: eruption present (1) or absent (0)[7]. Because the patterning dynamics of histamine concentration of mast cells and coagulation factors were identical (Supplementary Figs. S2–S6), we plotted the representative results as $[A](\mathbf{x}, t) \equiv [C](\mathbf{x}, t)$ in this study.

**Cell culture and impedance assay.** Endothelial cell growth supplement (ECGS), bovine serum albumin (BSA), and lipopolysaccharide (LPS) were from Sigma-Aldrich Japan (Tokyo, Japan). The human umbilical vein endothelial cells (HUVECs) were obtained from the American Type Culture Collection (ATCC, Manassas, VA, USA). Dulbecco's modified Eagle's medium (DMEM)/F12, penicillin, streptomycin, trypsin, and heparin were from Thermo Fisher Scientific (Waltham, MA, USA). Factor VIII-deficient human plasma and factor III (TF) were from COSMO BIO Co., Ltd. (Tokyo, Japan). Histamine was obtained from Wako Pure Chemical Industries, Ltd. (Osaka, Japan).

**Table 1 Criteria for classification of eruption geometry (EGe criteria).**

| Class | Clinical feature | Type | Clinical feature | |
|---|---|---|---|---|
| Boundary pattern | Comparatively flatted annular eruptions without surrounding erythema. | Annular pattern | Annular eruptions surrounded by a single arc with a sharp margin. They may be fused with each other. |  |
| | | Broken-annular pattern | Petal-like annular eruptions composed of multiple arcs. There is no continuity or intersection of individual arcs expanding centrifugally. |  |
| Area pattern | Inflated eruptions with/without flatted uniform erythema | Geographic pattern | Circular shaped wheals with a tendency to fuse with each other, forming a geographic appearance. |  |
| | | Circular pattern | Isolated circular wheals of ≥1 cm diameter with a low tendency to fuse together. |  |
| | | Dot pattern | Small wheals or erythema of <1 cm in diameter, with a low tendency to fuse together. |  |

Morphologically, eruptions in chronic spontaneous urticaria (CSU) are divided into two classes and five subtypes.

HUVECs were cultured in DMEM/F12 supplemented with 10% fetal calf serum (FCS), 100 U/mL penicillin, 100 μg/mL streptomycin, 40 μg/mL ECGS, and heparin.

For the impedance analysis, HUVECs were harvested using trypsin and seeded onto E-Plates (Roche Applied Science, Upper Bavaria, Germany) at a density of 50,000 cells/well. On day 2, cells were treated with LPS (100 ng/mL) and histamine (10 μM) only for the TF stimulation assay. On day 3, the E-plate was set on an iCElligence microelectronic biosensor system (Roche Applied Science) and impedance termed "cell index (CI)" was measured every 10 s. For the TF stimulation assay, cells were stimulated with indicated TF concentrations in DMEM/F12 with 2% BSA and 1% factor VIII-deficient human plasma without any supplements and FCS. For the histamine stimulation assay, cells were stimulated with indicated concentrations of histamine in DMEM/F12 supplemented with 10% FCS.

**Statistics and reproducibility.** Inhibition ratios of histamine release from human mast cells and basophils are shown as means with standard errors of three independent experiments. Gap formation of endothelial cells analyzed by impedance sensors is shown as a representative of three independent experiments. Statistical analysis was performed using Microsoft Excel™ (Version 16.78) in this study.

**Reporting summary.** Further information on research design is available in the Nature Portfolio Reporting Summary linked to this article.

## Results

**CSU mathematical modeling based on pathophysiological structures.** To design our CSU mathematical model, we first defined all the necessary pathophysiological molecules and their interactions that have been associated with CSU, based on in vitro experimental studies (Fig. 1). Firstly, we took into consideration that external stimuli or internal auto-allergens and auto-antibodies stimulate both basophils and mast cells to release mediators by crosslinking the high-affinity IgE receptors (FcεRI) on their surfaces[8–10] [Fig. 1a (0)]. Next, this process subsequently enhances TF expression on vascular endothelial cells [Fig. 1a (1)], which induces the extrinsic coagulation cascade[11–13] [Fig. 1a (2$_a$)]. Consequently, the coagulation factors increase the degranulation of basophils [Fig. 1a (2$_b$)] or skin mast cells via gap formation by vascular endothelial cells [Fig. 1a (2$_a$), (3), (4)] leading to edema [Fig. 1a (5), (6)][14]. Furthermore, adenosine, which is simultaneously released following degranulation of both human peripheral basophils and skin mast cells[15], suppresses histamine release and TF expression[16], we thus also took this into consideration (Fig. 1a, green inhibition symbols). Based on these facts, we applied the pathophysiological structure into a network structure (Fig. 1b), focusing on the following main factors that are expected to have a key role in the in vivo network: (i) concentration of histamine released from basophils ([$H_B$]), (ii) TF expression on vascular endothelial cells ([TF]), (iii) coagulation factors leaked from blood vessels ([$C$]), and (iv) histamine released from mast cells ([$H_M$]).

We developed the hierarchical mathematical model integrating both the intravascular and extravascular dynamics using in vitro

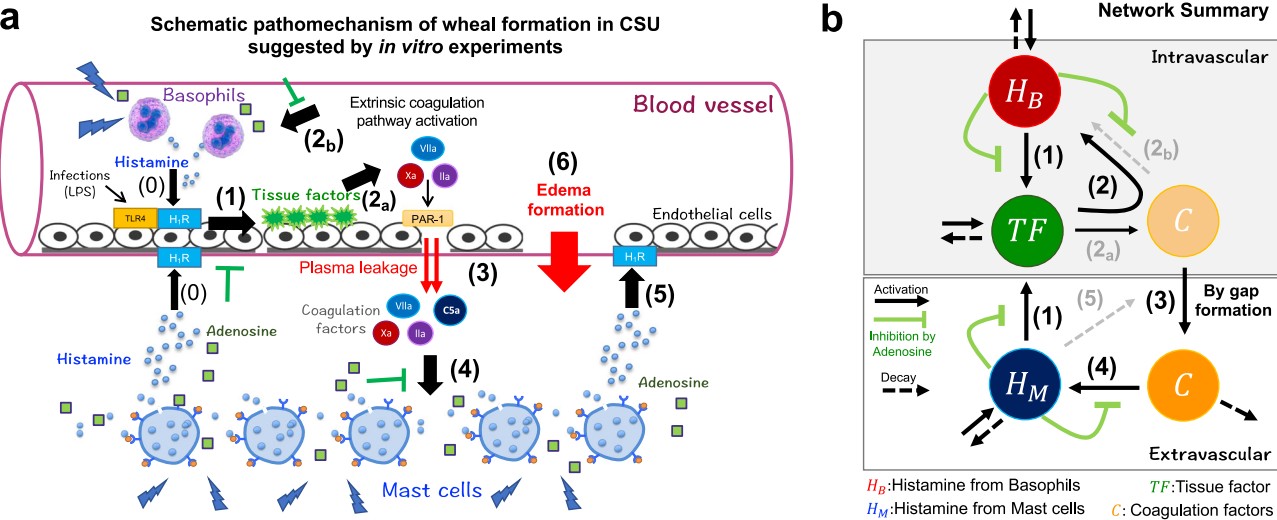

**Fig. 1 In vivo pathophysiological network of chronic spontaneous urticaria (CSU). a** Schematic diagram of CSU development predicted using in vitro experiments. Number labels correspond with those in (**b**). Black arrows and green symbols indicate positive (activation) and negative (suppression/ inhibition) regulation, respectively. The order of pathophysiological progression is sequentially numbered. **b** Summary of the network based on the description in (**a**). Networks that do not appear explicitly in the mathematical model are depicted by gray dotted arrows. In the mathematical model, arrows (2_a) and (2_b) were simplified to solid arrows (2). Networks (2_a) and (5) were integrated into the modeling of the gap-forming network (3). LPS lipopolysaccharide, H1R histamine H1 receptor, TLR4 toll-like receptor 4, PAR-1 protease activated receptor-1.

experimental data. Wheals in CSU are caused by dilatation and increased permeability of the postcapillary venules in the subpapillary layer of the dermis. The subpapillary layer of the dermis is the particularly shallow layer of the dermis with an average of 2 mm thickness, which depends on location. Compared with the scale that reproduces the horizontal spread of the wheals in the mathematical model (35 cm × 35 cm), the depth scale at which the vascular reactions involved in wheal formation is negligibly small. Thus, we mathematically assumed that the dermis of the skin can be considered as a thin two-dimensional (2D) sheet with numerous capillaries spread over it. We constructed a 2D spatial model with the intravascular space distributed homogeneously throughout the dermis. Based on the network (Fig. 1b), we constructed the following model (see Supplementary Methods 1–5 for details on model construction and parameter estimation).

$$\frac{d[H_B]}{dt} = \delta_B + \gamma_B[\text{TF}]\chi_B(\mathbf{x}, t)\big(1 - g^B_{\text{inhibition}}\big([H_B]\big)\big) - \mu_B[H_B]\,\text{on}\,\Omega_B, \quad (5)$$

$$\frac{d[\text{TF}]}{dt} = \delta_T + f^T_{\text{activation}}\big([H_B], [H_M]\big)\big(1 - g^T_{\text{inhibition}}\big([H_B], [H_M]\big)\big) - \mu_T[\text{TF}]\,\text{on}\,\Omega_E, \quad (6)$$

$$\frac{\partial[C]}{\partial t} = D_c\nabla^2 C + f_{\text{leakage}}(\mathbf{x}, t) - \mu_C[C]\,\text{on}\,\Omega_D, \quad (7)$$

$$\frac{\partial[H_M]}{\partial t} = D_M\nabla^2[H_M] + \delta_M + \gamma_M[C]\chi_M(\mathbf{x}, t)\big(1 - g^M_{\text{inhibition}}\big([H_M]\big)\big) - \mu_M[H_M]\,\text{on}\,\Omega_D, \quad (8)$$

where $\Omega_D(\subset \mathbb{R}^2)$ is the dermal region, $\Omega_E$ is the vascular endothelial tissue, and $\Omega_B(\subset \Omega_E)$ is a randomly chosen local region of vascular endothelial tissue where the basophils adhere to the endothelial cells. Furthermore, $\delta_B, \delta_T,$ and $\delta_M$ are the basal production rates and $\mu_B, \mu_T,$ and $\mu_M$ are the basal decay rates. $\gamma_B$ and $\gamma_M$ are histamine release rates of basophils and mast cells, respectively (network (2) and (4) in Fig. 1a, b). The terms $\chi_B(\mathbf{x}, t)$

and $\chi_M(\mathbf{x}, t)$ are the Heaviside step functions, assumed to be 1 when each basophil and mast cell contained histamine and 0 when they each released their total histamine content[7,17]. We quantitatively estimated the main network functions, $g^B_{\text{inhibition}}$, $g^T_{\text{inhibition}}, g^M_{\text{inhibition}}, f^T_{\text{activation}},$ and $f_{\text{leakage}}(\mathbf{x}, t)$ based on the in vitro experiment data.

$g^B_{\text{inhibition}}, g^T_{\text{inhibition}},$ and $g^M_{\text{inhibition}}$ represent the inhibition rates of adenosine which is produced by adenosine triphosphate (ATP) released simultaneously with and in proportion to histamine from basophils and mast cells[15] (green inhibition symbols in the networks shown in Fig. 1a, b). Adenosine is known to suppresses the histamine release of human peripheral basophils and human skin mast cells (hsMCs) in response to anti-IgE in a concentration-dependent manner[16]. Based on our previous data[16], we directly analyzed the inhibitory rates of histamine released from hsMCs and basophils via adenosine (Fig. 2a, b, respectively). Using these data, we estimated the specific forms of $g^B_{\text{inhibition}}$ and $g^M_{\text{inhibition}}$ as shown in Fig. 2c:

$$g^B_{\text{inhibition}}\big([H_B]\big) = \frac{\alpha_B[H_B]^2}{\alpha_{B0} + [H_B]^2}, \quad (9)$$

$$g^M_{\text{inhibition}}\big([H_M]\big) = \frac{\alpha_M[H_M]^2}{\alpha_{M0} + [H_M]^2}, \quad (10)$$

where $\alpha_B$ and $\alpha_M$ are the maximal inhibition rates, $\alpha_{B0}$ and $\alpha_{M0}$ are the constants that determine the degree of increase in the curves.

Next, using in vitro experiment data of the tissue factor (TF) suppression by adenosine[18], we also analyzed the inhibitory rates of TF mRNA expression based on adenosine concentrations (Fig. 2d) and estimated the inhibitory rate function of TF follows (Fig. 2e):

$$g^T_{\text{inhibition}}\big([H_B], [H_M]\big) = \frac{\alpha_T\big([H_B] + [H_M]\big)^2}{\alpha_{T0} + \big([H_B] + [H_M]\big)^2}, \quad (11)$$

where we estimated the parameter values, the maximal inhibition rate ($\alpha_T$) and a constant ($\alpha_{T0}$), by the least squares method.

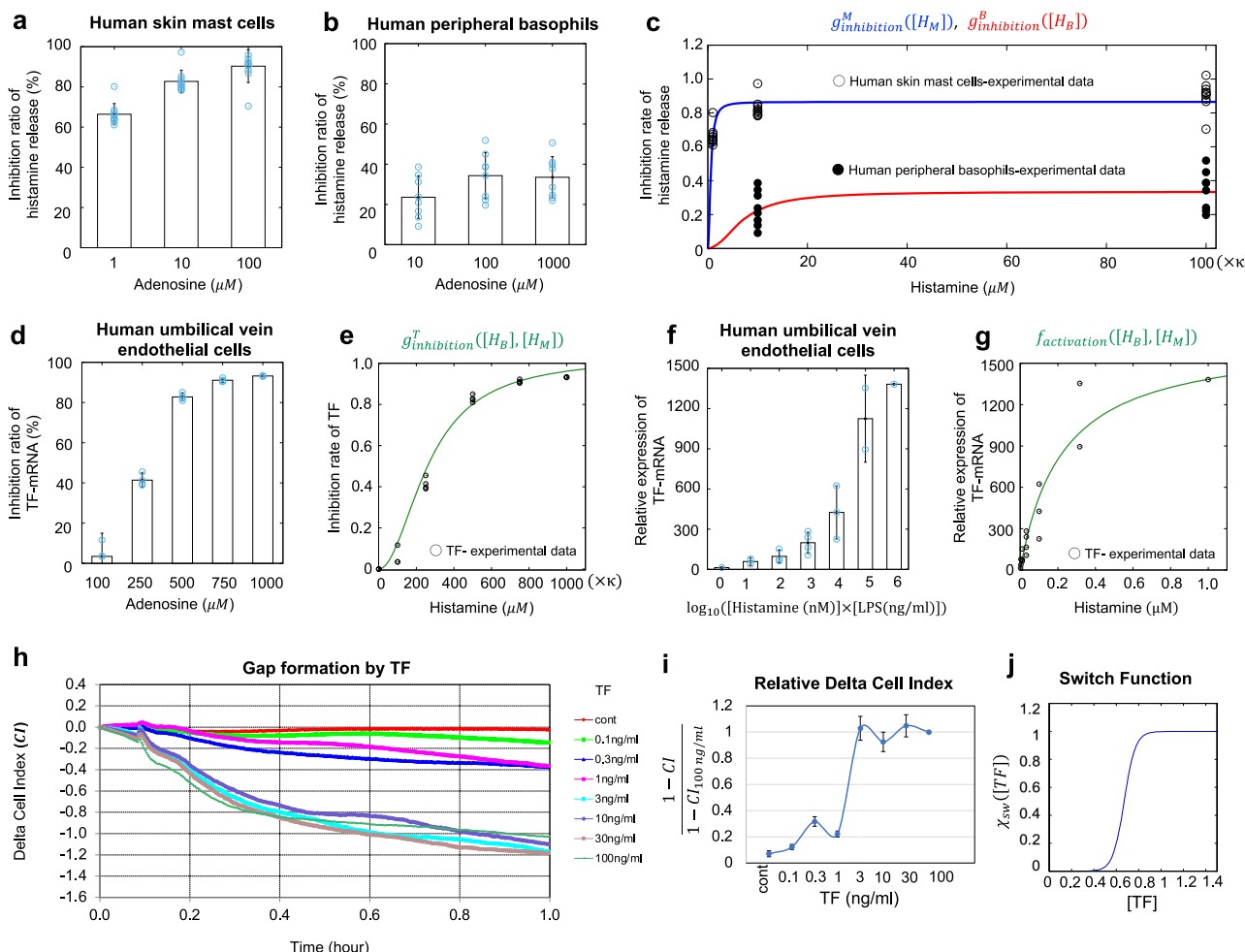

**Fig. 2 in vitro experiments for key networks and estimation of model functions. a**, **b** Inhibition ratio of histamine release from human skin mast cells (hsMCs)/human peripheral basophils in response to adenosine, calculated from data of[16]. **c** Inhibitory functions estimated from data of **a**, **b** with the data conversion of adenosine to histamine (Supplementary Method 2). Black and white circular points are the experimental data of human peripheral basophils and human skin mast cells, respectively. The blue and red lines are the fitted graph for each case. **d** Inhibition ratio of tissue factor (TF) mRNA from HUVECs induced by adenosine, calculated from experimental data of Yanase, Morioke, et al., 2018. **e** Inhibitory function estimated from the data of (**d**). **f** Relative expression of TF mRNA from HUVECs induced by co-stimulation with histamine and LPS. Each color corresponds to the TF concentration in the right panel. **g** Activation by TF estimated from data of (**f**). $\kappa$ is given in equation (s3) of Supplementary Method 3 and $\bar{\kappa}$ is a scaling parameter that converts LPS concentration to histamine concentration (details are presented in Supplementary Methods 2 and 3). **h** TF concentration-dependent gap formation. Temporal variation in cell index (CI) at different TF concentrations. Endothelial cells were stimulated with TF at indicated concentrations in Dulbecco's modified Eagle's Medium (DMEM)/F12 with 2% bovine serum albumin (BSA) and 1% factor VIII-deficient human plasma without any supplements and fetal calf serum (FCS). **i** Relative scale of CI, which was scaled by data obtained at 100 ng/mL each time. **j** Estimated switch function, $\chi_{sw}([\mathrm{TF}])$, obtained by quantitative analysis of variation in CI at different TF concentrations. In **a**, **b**, **d**, **f**, and **i**, the black point indicates the mean, and the error bars are standard deviations. The blue circles in **a**, **b**, **d**, and **f** are the data points.

$f_{\mathrm{activation}}^{T}$ represents a function of histamine-induced TF expression (networks (1) in Fig. 1a, b). TF in human umbilical vein endothelial cells can be induced by histamine and synergistically increased by co-stimulation with histamine and lipopolysaccharide (LPS)[18]. To reflect the potential synergistical output of TF activation in our model, we directly analyzed previous experimental data on the relative expression level of TF mRNA induced by co-stimulation with histamine and LPS (Fig. 2f). We then converted the result to data based on the histamine-only concentration and estimated the TF activation function as follows (Fig. 2g):

$$f_{\mathrm{activation}}^{T}\big([H_B],[H_M]\big) = \frac{\gamma_T\big([H_B]+[H_M]\big)}{\gamma_{T0}+[H_B]+[H_M]}, \qquad (12)$$

where $\gamma_T$ is the maximal activation rate and $\gamma_{T0}$ is a constant that determines the degree of increase in the curve.

The TF expressed on endothelial cells induces the extrinsic coagulation cascade[11,13], and activated coagulation factors produced by the extrinsic coagulation pathway induce intracellular gap formation between endothelial cells via PAR-1[19]. Thus, the coagulation factors leaked from intravascular, namely, $f_{\mathrm{leakage}}(\mathbf{x},t)$ can be determined by the dynamics of intercellular gap formation.

In order to determine the detailed form of $f_{\mathrm{leakage}}(\mathbf{x},t)$, we performed in vitro experiments using human umbilical vein endothelial cells (HUVECs). We looked for the changes in the adhesion area and morphology of the cells cultured on the electrodes with varying TF and histamine concentrations, respectively. We first measured the time-related cell index (CI) with varying TF by using an impedance analysis (Fig. 2h). We found that the CI was not significantly changed following treatment with TF at <1 ng/mL, whereas it was dramatically decreased with a TF concentration >3 ng/mL. Notably, the change

in CI was classified at two levels and its dynamics was switched by a threshold concentration of TF between 1 and 3 ng/mL. This result suggests that TF plays a role in inducing the leakage of coagulation factors via switch-like gap formation.

Based on this experimental result, we inferred the TF-dependent switch-like function, $\chi_{sw}([TF])$, by calculating the relative CI, $(1 - CI)/(1 - CI_{100ng/ml})$, which gives the relative variation in the size of the CI for each time point. We calculated the average relative CI at [0.2, 0.6] (hour) for each TF concentration and found the switch-like function as shown in Fig. 2I, j. Furthermore, we found that the rate of gap area expansion is determined in a concentration-independent manner for both TF and histamine (Supplementary Method 5, Fig. S1 in mathematical and experimental details). Taken together, we obtained

$$f_{\text{leakage}}(\mathbf{x}, t) = \frac{\gamma_C}{1 + \exp\left[-\beta\left([TF] - T_{sw}\right)\right]}, \quad (13)$$

where $\gamma_C$ is a leaked rate of coagulation factors (Fig. 2j).

At this stage, we had completed the CSU mathematical model (5)–(8) based on experimental analysis of the key pathophysiological networks. The eruptions of CSU visibly appeared on the skin where the leaked coagulation factors from blood vessels or histamine concentration were high in the dermis. Therefore, we plotted the eruption state of the skin using the eruption state function (defined in the Methods).

**Recapitulation of eruption patterns in urticaria using a mathematical model**. We next validated whether our mathematical model could successfully recapitulate the CSU eruption patterns. Depending on the parameter values, we identified several eruption patterns observed in the patients with urticaria (Fig. 3a–e, Figs. S2–S6). Based on geometric characteristics, we categorized the different eruption patterns identified in our mathematical model as follows: annular, broken-annular, geographic, circular, and dot patterns.

The eruption image data of patients in clinical medicine represent the data of a certain time point of eruption development dynamics. Thus, we compared the in silico eruption patterns at some fixed time point with the image data collected for CSU patients who visited the hospital (Fig. 3f). We found that patterns among the patient data were highly similar to all pattern types observed in the in silico experiments, hence validating our CSU mathematical model.

**Classification of eruption geometry in urticaria patients according to eruption geometry criteria**. Interestingly, during the in silico validation of our mathematical model, we observed that eruption patterns were typically identified as one of five types when parameter values were varied (Fig. 3g, h). Therefore, we next sought to determine whether all eruption patterns appearing in CSU patients could be classified into specific types and then linked to in vivo pathologies. To this end, we first developed classification criteria defining the clinical features of the eruption (wheal) geometry found in the in silico patterns (Table 1). We designated the resultant table of clinical features *Criteria for Classification of Eruption Geometry* (EGe criteria). We classified the patient eruption patterns into two large classes, boundary and area patterns, which we then further classified into five specific patterns, namely annular, broken-annular, geographic, circular, and dot patterns, as observed in the in silico experiments.

To confirm whether the eruption patterns of actual CSU patients could be classified into one of the eruption types in the EGe criteria, we collected the eruption image data of 105 patients who visited the Hiroshima University Hospital with CSU-related skin eruptions (see Online Method). The data were then analyzed

by six dermatologists, based on the EGe criteria. We evaluated the reliability **SB** ($\equiv$[number of doctors who selected the sample as one of the types in the EGe criteria]/[number of dermatologists (6)]), of the statistical result using the vote-counting method.

The results showed that 87.6% of the samples could be classified as one of the eruption types with 100% **SB** reliability (Fig. 4a), and 0% of the samples were unclassifiable, with a reliability **SB** > 83% (Fig. 4b). We also statistically confirmed that the classifiable rate of 87.6% is not a random chance by the *P*-value (Supplementary Method 6). This result proves that the eruption types in CSU patients could be classified into one of the patterns of the EGe criteria developed based on our mathematical model. This suggests that the eruption dynamics of CSU patients can be determined using our CSU mathematical model.

Next, we investigated which types of eruption patterns were commonly observed in CSU patients. To obtain results with higher reliability, we introduced a new measurement of reliability (**SS**) defined as **SS** = [**SB** of type selection] × [**SB** of class selection] (Fig. 4c, Online Methods). From the analyzed data with the reliability **SS**, we found that geographic and circular types were observed at high frequencies, whereas boundary patterns appeared fewer times than area patterns did.

**Linking the key features of eruption geometry to related pathological networks via a mathematical model**. We next extracted the minimal sets of key characteristics of eruption patterns from the features defined in the EGe criteria (Table 2): distinguishable (KF1)/indistinguishable (KF3) between the boundary and interior of the wheal, disconnectedness in the wheal boundary (KF2), fusion of the wheals (KF4), and no change in size of wheals (KF5). This showed that each of the eruption patterns could be identified using appropriate combinations of the five features. Specifically, the boundary pattern is identified by KF1, the annular/broken-annular pattern is identified by KF1 and KF2, the geographic/circular pattern is identified by KF3 and KF4, and the dot pattern is identified by KF3 and KF5.

We then looked at how each eruption pattern is connected to the network of pathological dynamics. To this purpose, we performed a global sensitivity analysis of the Extended Fourier Amplitude Sensitivity Test (eFAST) ([20]; explained in detail in Supplementary Method 7), which tells us which networks play an important role for a given sensitivity function, namely, each eruption pattern. Based on the five key features of clinically observed wheals in CSU (Table 2), we defined five sensitivity functions to measure the degree of alteration of the features of each eruption pattern in response to changes in parameter values representing the network intensity (Fig. 5, Supplementary Method 7 shows the details of sensitivity functions).

We first noted that most key features had a relatively high association with the networks related to TF and mast cells (Fig. 5f, yellow-colored regions), suggesting that the dynamics of TF and mast cells may be important players in determining the geometrical properties of CSU eruptions. We next compared the difference between boundary and area features. In the boundary feature (KF1), the decay rate of TF ($\mu_T$) highly influenced the features (Fig. 5a, f). By contrast, we found that $\mu_T$ had a negligible effect on the area feature (KF3) (Fig. 5c, f). In addition, the activation network of TF ($\gamma_T$ and $\delta_T$) had a considerable effect on the area feature but not on the boundary feature, although an effect was evident (Fig. 5f). These results suggest that the feature of boundary/area pattern is likely to be highly influenced by TF degradation/activation.

Next, to determine the network that played a critical role in the divergence of the annular and broken annual patterns, we compared the sensitive analysis result between the boundary

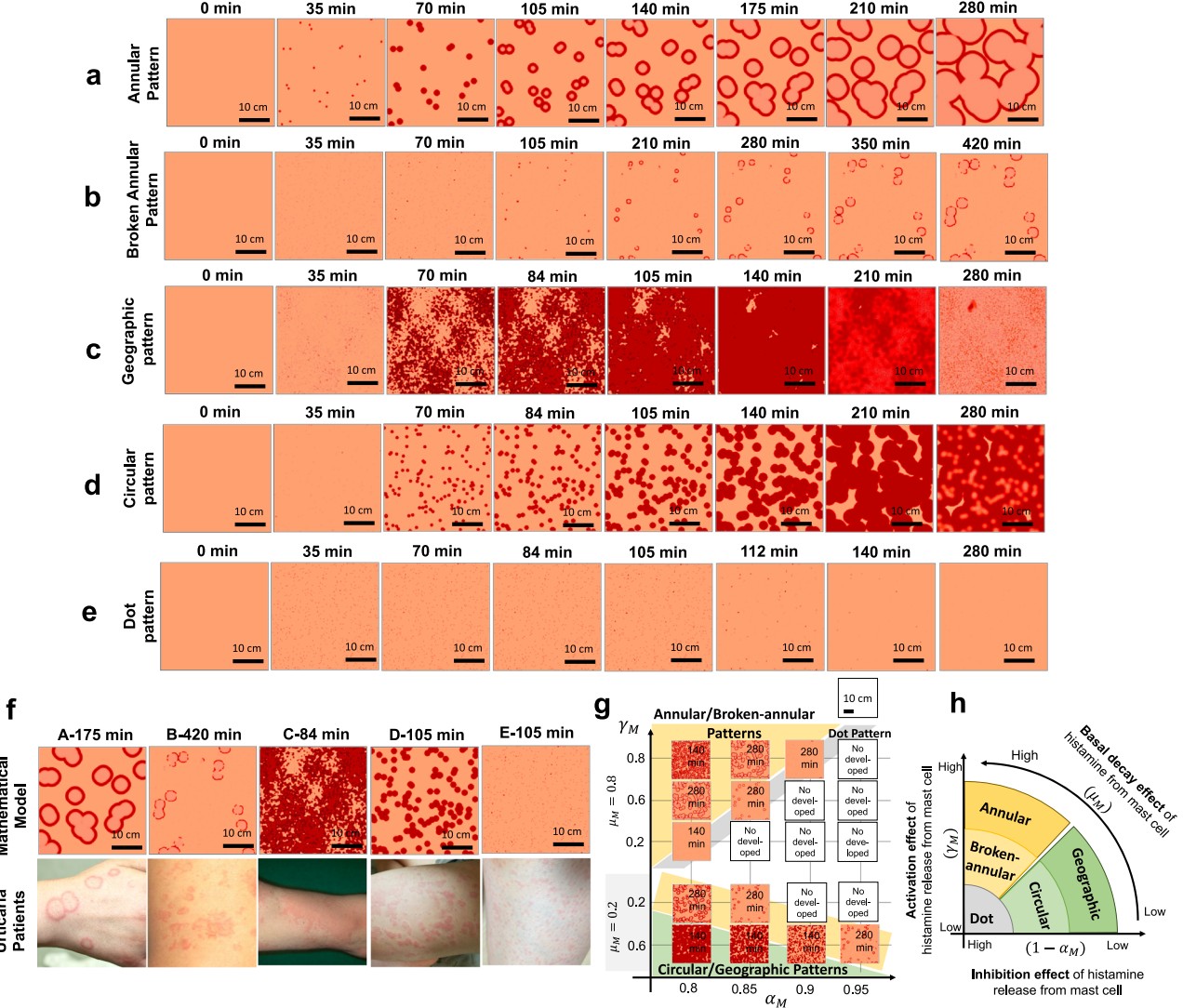

**Fig. 3 Five wheal patterns identified using a mathematical model. a–e** Evolution of each wheal pattern. **f** Comparison of wheal pattern using a mathematical model at a time point with that of urticaria patients. Details of parameter values are shown in Table S1. **g** The panel shows eruption patterns by changing parameter spaces: activation of histamine release ($\gamma_M$), decay rate of histamine ($\mu_M$), and inhibition rate of histamine ($\alpha_M$) from mast cells. Yellow regions indicate the parameter region in which annular/broken-annular patterns appear. Green regions indicate the parameter region in circular/geographic patterns. Gray regions indicate the parameter region in dot patterns. **h** A schematic diagram of the eruption pattern based on three parameters.

(KF1) and boundary disconnectedness (KF2) features (Fig. 5a, b). We found that the effect of the basal activation network ($\delta_M$) of histamine release from mast cells in the disconnectedness feature (KF2) was not as high as that of the boundary feature (KF1), indicating that the state of spontaneous histamine release from mast cells may be the critical factor to make the divergence of the annular and broken annual patterns.

We then compared the area fusion (KF4) and punctate (KF5) features (Fig. 5d, e). The first notable differences between these two features were the activating effect of spontaneous histamine release from mast cells($\delta_M$) and the inhibitory effect of histamine release from mast cells($\alpha_M$). Both the networks ($\delta_M$ and $\alpha_M$) showed a high effect on the area fusion feature, whereas they exhibited relatively negligible effects on the punctate feature. This result suggests that the area types of eruption patterns may become geographic/circular type or dot type depending on whether the histamine dynamics from mast cells are strongly involved or not. We summarized the classification of eruption types and their relations with TF and histamine dynamics of mast cells based on the sensitive analysis in Fig. 5g.

## Discussion

In this study, we established a novel approach to overcome the limitations of clinical data by developing a three-step research strategy: generating a mathematical model that recapitulates eruption patterns from the pathological structures (Fig. 5h, Step 1), establishing EGe criteria for the clinical classification of eruptions in patients based on in silico eruption patterns (Fig. 5h, Step 2), and linking the features of eruption geometry to critical networks of pathological structures (Fig. 5h, Step 3). With this stepwise approach, we determine the in vivo pathological molecular dynamics in urticaria patients via their eruption geometry, by extracting the geometric features of skin eruptions from patients with urticaria and linking these features to the parameter values of the CSU mathematical model.

In the first step, we successfully developed a hierarchical urticaria mathematical model in which all the model equations were verified by in vitro experiments. To our knowledge, this is the first study wherein the pathophysiological steps involved in the development of CSU were structurally proven in silico. The first conceptual mathematical model of urticaria was proposed in our

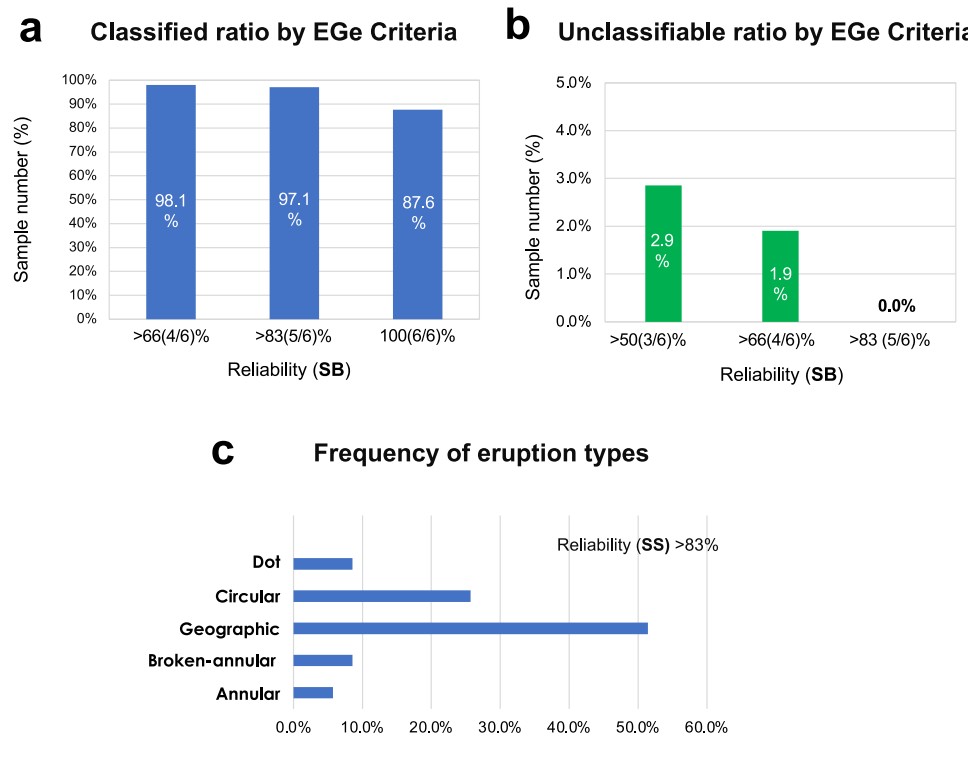

**Fig. 4 Classification of eruption patterns by the eruption geometry criteria (EGe criteria) for CSU patients. a** Ratio of patient samples in which six dermatologists selected one of the pattern types defined by the EGe criteria. **b** Ratio of patient samples selected as not classifiable by the EGe criteria. Data were calculated based on reliability **SB**. Biologically independent patient samples ($n = 105$) were analyzed in (**a**) and (**b**). **c** Frequency of patient eruption types. Data ($n = 35$ biologically independent patient samples) were selected among $n = 105$ biologically independent patient samples with reliability **SS** > 83%.

**Table 2 Summary table of eruption geometries based on key features extracted from EGe criteria.**

|  | Boundary pattern | | Area pattern | | |
|---|---|---|---|---|---|
| **Key features (KF)** | **Annular** | **Broken-annular** | **Geographic** | **Circular** | **Dot** |
| (KF1: Boundary) Distinguishable between wheal boundary and interior | ✓ | ✓ |  |  |  |
| (KF2: Boundary-disconnected) No connectivity of wheal boundary |  | ✓ |  |  |  |
| (KF3: Area) Indistinguishable between wheal boundary and interior |  |  | ✓ | ✓ | ✓ |
| (KF4: Area fusion) Increase of wheal area size |  |  | ✓ |  |  |
| (KF5: Punctate) Small and no change in size over time |  |  |  |  | ✓ |

The table below provides a simple means of classifying eruption geometry based on five key features.

recent work[7]. In the previous model, we proposed that the inhibitory regulation of histamine release from mast cells plays an important role in creating multifarious patterns of the wheal geometry, although this finding has not been experimentally verified. In this study, the inhibition dynamics of adenosine were modeled based on the experimental data[16] and proved that the inhibitory regulation in mast cells is likely to be involved in the classification of eruption types with interactions in other physiological networks (Fig. 4g, Fig. S7). Moreover, our CSU model proposes that the hierarchical structure of urticaria development as well as the key players, such as TF and histamine released from mast cells, are critical in generating diverse eruption patterns and determining the eruption types.

In the process of developing the mathematical model, we also found that the intracellular gap in endothelial cells of the blood vessels is formed at a threshold level that depends on TF concentration. This finding suggests that wheals may develop only when stimuli are presented at a level above a certain threshold.

Thus, the gap formation dynamics based on the TF switch may affect not only the formation of eruption patterns but also the onset of wheal formation.

Using our model, we also identified five eruption patterns and created EGe criteria for the classification of urticaria eruptions in real patients. Application of the EGe criteria confirmed that patients with CSU could be classified into one of the five types, and the types included in the area pattern were observed more frequently than those in the boundary pattern in the patients. Authors experience boundary patterns specifically in patients with spontaneous urticaria, whereas area patterns in both spontaneous and inducible types of urticaria (data not shown). Therefore, area patterning may reflect the ubiquitous molecular mechanisms common to patients and various types of urticaria, thereby resulting in a high patterning frequency. Taken together, these results indicate that our EGe criteria could be a tool of practical approach for clinicians to apply in determining eruption patterns of patients in the future.

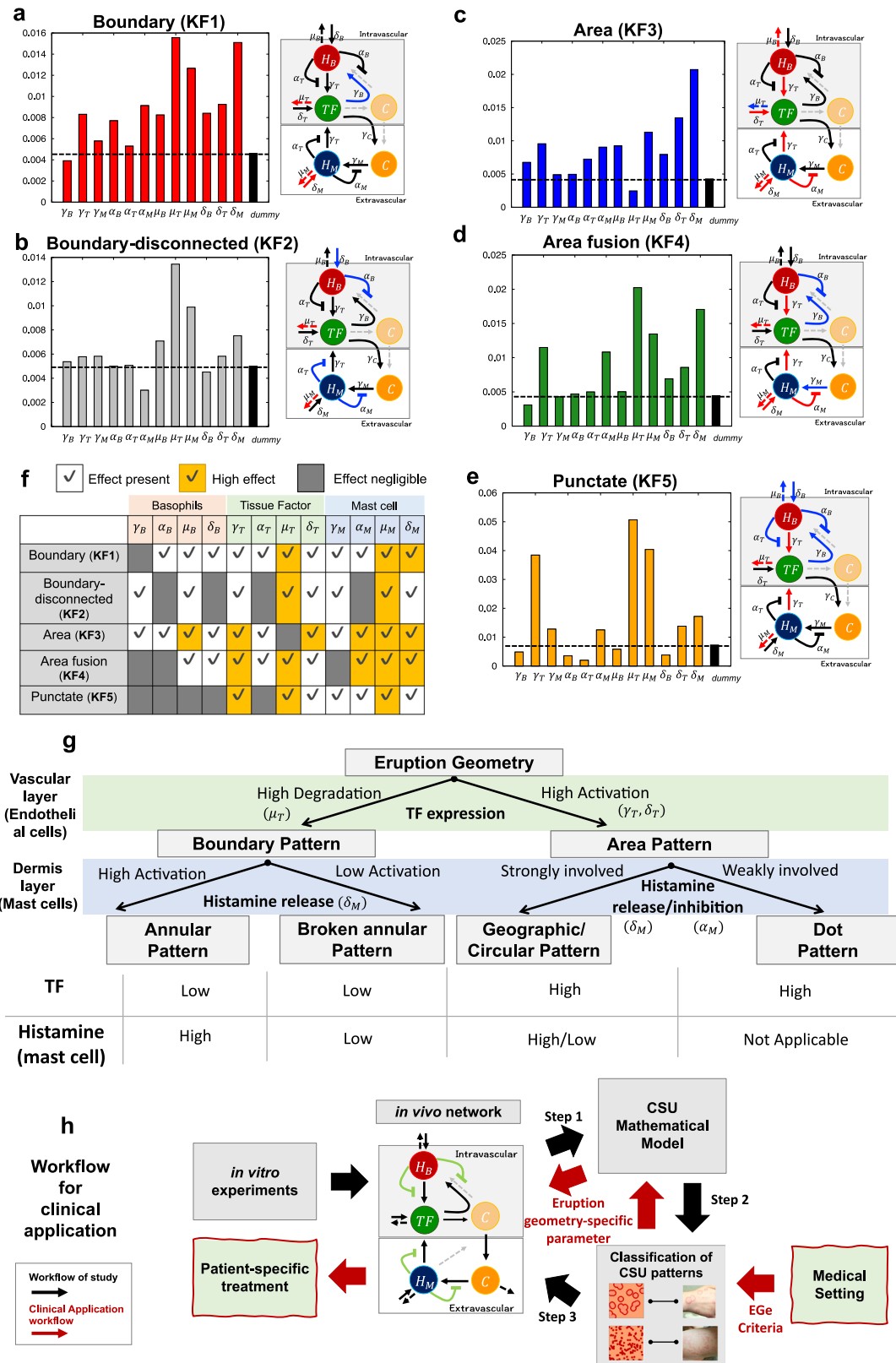

In the final step of our research strategy, we inferred the pathophysiological networks that are critically involved in each feature of the eruption geometry. We found that the balance of effects by TF and the histamine released from mast cells play a critical role in determining the types of eruption patterns. Our result suggests that the level of TF expression may be involved in an underlying mechanism mediating the creation of notable differences in eruption shapes such as boundary or area patterns. Furthermore, the kinetics of histamine released from mast cells may also play a role in determining the specific type of eruption in different patients. This implies that even suppressing the action of a specific network using a drug may not always exhibit the same effectiveness among patients exhibiting different patterns of eruptions. In fact, antihistamines are effective for 70–80% of CSU

**Fig. 5 Linking key features of wheal geometry to related networks. a–e** Results of the first index determined using extended Fourier amplitude sensitivity test (eFAST) sensitive analysis of each feature. The first-order index of the dummy parameter was calculated to confirm that index values in panels (**a–e**) yielded reliable data to determine the significance of model parameters[28]. For model parameters with an index less than or equal to that of the dummy parameter, the parameters are considered not significantly different from zero. Networks with indexes lower than (negligible effect) and double the height of the dummy parameter (high effect) are indicated in blue and red, respectively. **f** Summary table of sensitivity analysis. Parameter sets related to basophil, TF, and mast cell dynamics are shaded by the same color. **g** Summary of analysis results of the pattern types and main network. The table at the bottom shows relative levels of tissue factor (TF) expression (green shaded region in the upper panel) and histamine released from mast cells (blue shaded region in the upper panel) for each eruption pattern. (**h**) Summary of workflow and clinical application. Black arrows show the study workflow and red arrows represent the general clinical applications of our research, leading to potential patient-specific treatments for intractable skin diseases.

patients[21], but certain patients refractory to antihistamines respond well to anticoagulants and protease inhibitors[22–24]. Thus, our approach, which proposes a pathophysiological in vivo network considering both topological and kinetical reactions in the skin, is expected to suggest a patient-specific treatment based on the geometry of skin eruptions (Fig. 5h, red arrows).

In the model assumed in this study, histamine plays an important role in a cascade activating mast cells. Our previous studies suggest that histamine may play both in an early step in inducing tissue factor expression on vascular endothelial cells and a late step in inducing plasma extravasation via the increase of vascular permeability (Fig. 1)[25]. However, its role in the cascade, especially in the early step, may be substituted to some extent by other factors, such as LPS and cytokines. Therefore, the inhibition of histamine by antihistamines in an early step of the cascade may or may not fully prevent the process of consequential steps involving the coagulation pathway, depending on the degrees of the involvement of histamine in this step in individual patients. In this study, we also revealed the association between high TF activation and the area pattern rather than boundary ones by theoretical analysis. Such an association may be experimentally confirmed in future studies by examining levels of coagulation biomarkers such as PF1 + 2 and D-dimer[26,27]. Thus, the relation of wheal pattern to the biomarkers of coagulation as well as autoimmunity activating mast cells and basophils needs to be evaluated in the future.

Urticaria involves both temporal and spatial dynamics, and the eruptions show highly dynamic transitions from the phases of eruption emergence (onset), development, and disappearance. The distinction between annular and broken-annular patterns, and that between circular and geographic patterns may be sometimes difficult in certain clinical practices. However, the five wheals' patterns are clearly distinguished in the course of mathematical topology (Supplementary Table S2). The wheal expands of annular patterns keep the connected boundary, whereas those of broken-annular develop with disconnected boundaries (Fig. 3). Such difference should be more apparent in kinetics information rather than a snapshot image of individual wheals. This feature is also clearly distinguishable in mathematical topology. Similarly, the features between circular and geographic patterns will be more clearly distinguished in the temporal dynamics of wheal fusion. In in silico case, we characterized the circular pattern of wheal by slow speed of expansion and sparse distribution of the emerging points. Thus, time-lapse views of wheals are expected to be obtained for analysis and endorse the model proposed in this study in future studies.

This study is limited to focusing on the emergence and development phases, and we did not consider the disappearance phase. In the current model, the eruptions would continue to expand if the simulation space is not limited. This suggests that the disappearance phase may be caused by a network independent of the onset phase. Understanding the overall dynamics of CSU from the onset to the disappearance phase will be crucial to understanding the exact pathological state of patients based on

the eruption state. To achieve this, the time-related historical data of patients will need to be acquired and added to our current mathematical model. Furthermore, we constructed a mathematical model based on in vitro experimental data and succeeded in explaining the characteristics of the eruption geometry. However, a more accurate clinical and biological validation using another independent data set is needed.

Despite these limitations, the findings of this study suggest that the skin eruption geometry is linked to the dynamics of in vivo molecular networks involved in the pathogenesis of CSU. The inference of the detailed parameter values in our mathematical system for each patient may be applied to decide patient-specific treatments. Moreover, the mathematical modeling approach used in this study may be applicable not only to urticaria but also to a variety of skin diseases characterized by the formation of geometric morphology, such as psoriasis, tinea corporis, erythema multiforme, autoimmune bullous disease, erythema gyratum repens. Thus, our multidisciplinary approach linking the skin eruption geometry and in vivo, dynamics could help develop a novel strategy for dermatological research to manage intractable skin diseases in an integrated manner.

## Data availability

Supplementary Data 1 contains source data for the main figures (Fig. 2 and Fig. 4) with numerical results in this paper. The dataset was obtained as part of a clinical study (Ethical Committee for Epidemiology of Hiroshima University, Hiroshima, Japan, approval number; E2020-2388). The dataset of sensitivity analysis is available from the corresponding author (SSL) upon reasonable request.

## Code availability

Numerical code is made available online on GitHub and all other data are available from the corresponding author (SSL) on reasonable request.

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

## Acknowledgements

This study was mainly supported by the Japan Science and Technology Agency (JST) CREST (JPMJCR2111) to S.S.L. and partially funded by grants to Y.Y. from MEXT Grant-in-Aid for Scientific Research (21K08301) and to M.H. from MEXT Grant-in-Aid for Scientific Research (21K08346). We thank Drs. Akiko Kamegashira, Satoshi Morioke, and Akio Tanaka at Hiroshima University Hospital for their performing the classification of the morphology of wheals in the clinical part of the study. We thank Dr. Imori of Hiroshima University for their valuable comments on statistical analysis.

## Author contributions

S.S.L., S.T., and M.H. initiated and designed the research. S.S.L. executed the research and analyzed data. S.S.L. developed a mathematical model and numerical algorithms for simulations. S.S.L., D.M., Y.Y., S.T., and M.H. supervised the data collection and provided the data. Y.Y. designed the experiment and analyzed the data. T.K. executed the experiment. S.S.L. carried out data analysis for experimental and clinical data. S.S.L. wrote the first draft. S.S.L., S.T., Y.Y., and M.H. revised the paper.

## Competing interests

S.T. has received a research grant from Sanofi, Maruho, Tanabe-Mitsubishi, Eli Lilly, and Taiho Pharmaceutical, honorarium from Tanabe-Mitsubishi, Taiho Pharmaceutical, C.S.L. Behring, Kaken, Maruho, and Abbvie, advisory fee from Sanofi, and the gift of a medication for a clinical study from Takeda. M.H. has been a speaker or has served on advisory boards for Kaken Pharmaceutical, Kyowa Kirin, Kyorin, Mitsubishi Tanabe Pharma, Novartis, Sanofi, TAIHO Pharmaceutical, Takeda, Teikoku Seiyaku, and Tori. The remaining authors declare no competing interests.
