## [Peer Review File · Communications Medicine]

Reviewers' comments:

Reviewer #1 (Remarks to the Author):

In this paper, using a novel approach, Seirin-Lee et al show that the pathological state of CSU can be inferred by geometric features of the skin eruptions. Their hierarchical mathematical modelling approach is unique and the analysis of 105 real CSU patients achieved 87.6% agreement by dermatologists, which may be useful for the clinical understanding of CSU.

There are several points and limitations to be cleared.

The authors' assumption of a two-dimensional sheet with numerous capillaries spread over the dermis apparently corresponds to only superficial vascular plexus ignoring deep vascular plexus as well as its communicating vessels. This should be appropriately described.

Regarding the 5 wheal patterns I cannot figure out the mechanical difference between annular pattern and broken-annular pattern, which should be pathologically similar, because annular pattern would easily be converted to broken annular pattern. Similarly circular pattern would turn to geometric pattern (see Fig. 3). These overlapping could be observed in the same patient at the same time in different body sites. These points should be discussed more thoroughly.

The mathematical model is apparently well constructed and fitted to the clinical morphology. I admit it is remarkable that their hypothetical assumption could nicely explain clinical presentation. However, it should be emphasized that the model is only 'explanatory' per se for the authors' present cases and the validation should be performed using another independent data set.

Minor point

Among the critical parameters, (if I understand correctly), the authors do not distinguish TF formation (mRNA) and its release from the cell surface (see for example, Fig.2). These are clearly different and should be more precisely discussed.

There are several duplicated typing errors (such as page 8: line 27 and page 10: line7)

Reviewer #2 (Remarks to the Author):

This manuscript reports the results of a mathematical model based on 105 clinical pictures of chronic urticaria supplemented with laboratory data. The manuscript is well written and the data presented is new and interesting. It may lay base for an in silico research in chronic urticaria. It hardly contains any clinical and only some laboratory data and is mainly based on complex computer modelling.

What may be of special interest are the EG criteria that may be useful also in future clinical studies.

In my personal clinical experience, patients develop all recommended patterns of urticaria and not a patient specific pattern of their urticaria. This is different from other dermatological diseases such as polymorphous light eruption where the eruptions are polymorphous between different patients but mostly monomorphous within a single patient. How do the authors think about this aspect?

If other factors than histamine play an important role in the model of the authors, why do simple antihistamines sufficiently suppress all symptoms in up to 80 % of the patients with non-severe urticaria as mentioned in Line 329?

Reviewer #3 (Remarks to the Author):

The paper proves to be very interesting. The proposed hierarchical mathematical model could constitute an innovative approach to the clinical analysis of an intractable disease such as Chronic spontaneous urticaria. The classification of the eruptions in 5 distinct categories (with attention to the characteristic immunological signatures) can represent a good point of view to address the problem.

The clinical setting can be studied in depth thanks to the by geometric features of the eruptions, but one must be very careful because, precisely by virtue of the lack of experiments on model animals and the poor knowledge of the pathogenic mechanisms of the disease, the computational approach is a useful hypothesis which, in my humble opinion, needs to be considered but, at the moment, validated by more in-depth research. The reported computation is in any case appropriate and this paper, therefore, leads to further insights and deserves to be published.

This is our Response file to referees' reports in *Communications Medicine*.

Response to referees reports for manuscript COMMSMED-23-0076-T:

We thank the reviewers for the insightful comments and suggestions. We have detailed our responses below in **black**, with reviewer remarks in **green**. Changes in the revision are highlighted in **blue texts**, including the correction of additional typographical errors.

Reviewer #1: In this paper, using a novel approach, Seirin-Lee et al show that the pathological state of CSU can be inferred by geometric features of the skin eruptions. Their hierarchical mathematical modelling approach is unique and the analysis of 105 real CSU patients achieved 87.6% agreement by dermatologists, which may be useful for the clinical understanding of CSU. There are several points and limitations to be cleared.

We thank the referee's valuable and prospective comments, indeed. We answer to comments one by one in the following.

The authors' assumption of a two-dimensional sheet with numerous capillaries spread over the dermis apparently corresponds to only superficial vascular plexus ignoring deep vascular plexus as well as its communicating vessels. This should be appropriately described.

We thank the referee's careful indication. Medically, wheal formation in CSU is caused by the dilatation and increased permeability of the postcapillary venules in the subpapillary dermis. The subpapillary dermis is the superficial layer of the dermis in average of 2 mm thickness. Compared with the horizontal scale of the wheal formation in the mathematical model (35 x 35 cm), the vertical scale of depth at which the vascular reaction occurs is negligibly small. Thus, mathematically we assumed that the depth of dermis is sufficiently small and can be considered as 2-dimensional sheet. We added this point to the result section of mathematical model development (p. 4).

Regarding the 5 wheal patterns I cannot figure out the mechanical difference between annular pattern and broken-annular pattern, which should be pathologically similar, because annular pattern would easily be converted to broken annular pattern. Similarly circular pattern would turn to geometric pattern (see Fig. 3). These overlapping could be observed in the same patient at the same time in different body sites. These points should be discussed more thoroughly.

Thank you for your suggestion. The shape and size of wheals in a patient with urticaria change in time course. Moreover, their morphological patterns may also change in a course from the emergence to the diminishment. We agree that basic mechanism of wheal formation in CSU may be the same in all five patterns proposed in this study. Indeed, we produced five patterns of wheal in silico, using the same set of formulae with various quantitative parameters. Nevertheless, clinical experiences of the authors suggest that wheals developed in individual patients may be characterized by a certain pattern of the shape and kinetic. In this study, five pattern of wheal shapes have been mathematically generated in silico, and we confirmed that wheals observed in 105 patients with CSU may be well assigned according to this classification. We also agree that distinction between annular and broken-annular pattern, and that between circular and geographic patterns may be sometimes difficult in certain clinical practices. However, the five wheals patterns are clearly distinguished in time-course of mathematical topology. The wheal expands of annular pattern keep the connected boundary, whereas those of broken-annular develop with disconnected boundary (Fig. 3). Such difference should be more apparent in kinetics information rather than a snap shot image of individual wheals. The difference between circular and geographic patterns will also be more clearly distinguished in dynamics of wheal fusion. Mathematically, we characterized the circular pattern of wheal by slow speed of expansion and sparse distribution of the emerging points. Thus, time-lapse views of wheals are expected to be obtained for analysis and endorse the model proposed in this study in future study. We added this point in the Discussion section (p.11) and added a new table describing mathematical feature of eruption pattern corresponding to clinical features of EGe Criteria in Supplementary Information (Table S2 (p.20)).

The mathematical model is apparently well constructed and fitted to the clinical morphology. I admit it is remarkable that their hypothetical assumption could nicely explain clinical presentation. However, it should be emphasized that the model is only 'explanatory' per se for the authors' present cases and the validation should be performed using another independent data set.

We really thank the referee's understanding for the potential of mathematical modeling. We do agree with the referee's opinion and added the limitation of current mathematical model in the Discussion section (p.12).

Minor point

Among the critical parameters, (if I understand correctly), the authors do not distinguish TF formation (mRNA) and its release from the cell surface (see for example, Fig.2). These are clearly different and should be more precisely discussed.

TF is mainly expressed on the surface of cell membrane. The expression rate of TF on the cell surface was assumed to be linearly dependent with the amount of mRNA in a mathematical model. Some of them are released on the surface of microparticles. However, microparticles released from the surface of vascular endothelial cells should be immediately diffused in the blood circulation, and, therefore, their contribution should be low, if any, compared with TF on vascular endothelial cells. We added this point precisely in the mathematical modeling section of Supplementary Information (p. 2).

There are several duplicated typing errors (such as page 8: line 27 and page 10: line7)

We thank the referee's careful reading. We fixed all.

Reviewer #2:

This manuscript reports the results of a mathematical model based on 105 clinical pictures of chronic urticaria supplemented with laboratory data. The manuscript is well written and the data presented is new and interesting. It may lay base for an in silico research in chronic urticaria. It hardly contains any clinical and only some laboratory data and is mainly based on complex computer modelling. What may be of special interest are the EG criteria that may be useful also in future clinical studies.

We thank the referee's valuable comment.

In my personal clinical experience, patients develop all recommended patterns of urticaria and not a patient specific pattern of their urticaria. This is different from other dermatological diseases such as polymorphous light eruption where the eruptions are polymorphous between different patients but mostly monomorphous within a single patient. How do the authors think about this aspect?

Thank you for pointing out an important point. We think that mostly monomorphous eruptions within a single patient suggest that a potent single, and presumably endogenous, mechanism dominates the pathogenesis of all eruptions in the patient, whereas the lack of common patten(s) of eruptions among patients is due to the involvement of a variety of endogenous pathomechanisms despite exogenously single or a common mechanism, such as sun-light exposure in polymorphous light eruption. We retain this discussion within the review process of this manuscript for future studies, because it is too far out of scope of this manuscript.

If other factors than histamine play an important role in the model of the authors, why do simple antihistamines sufficiently suppress all symptoms in up to 80 % of the patients with non-severe urticaria as mentioned in Line 329?

In the model assumed in this study, histamine plays an important role in a cascade activating mast cells. Our previous studies suggest that histamine may play both in an early step to induce tissue factor expression on vascular endothelial cells and a late step to induce plasma extravasation via the increase of vascular permeability (Fig.1, Yanase Y, et al. Basophil Characteristics as a Marker of the Pathogenesis of Chronic Spontaneous Urticaria in Relation to the Coagulation and Complement Systems. *International Journal of Molecular Sciences* 2023, 24(12)). However, its role in the cascade, especially in the early step, may be substituted to some extent by other factors, such as LPS and cytokines. Therefore, the inhibition of histamine by antihistamines in early step of the cascade may or may not fully prevent the process of consequential steps involving the coagulation pathway, depending on degrees of the involvement of histamine in individual patients. We added this discussion in the Discussion (P.11).

Reviewer #3:

The paper proves to be very interesting. The proposed hierarchical mathematical model could constitute an innovative approach to the clinical analysis of an intractable disease such as Chronic spontaneous urticaria. The classification of the eruptions in 5 distinct categories (with attention to the characteristic immunological signatures) can represent a good point of view to address the problem. The clinical setting can be studied in depth thanks to the by geometric features of the eruptions, but one must be very careful because, precisely by virtue of the lack of experiments on model animals and the poor knowledge of the pathogenic mechanisms of the disease, the computational approach is a useful hypothesis which, in my humble opinion, needs to be considered but, at the moment, validated by more in-depth research. The reported computation is in any case appropriate and this paper, therefore, leads to further insights and deserves to be published.

We thank the referee very much for the constructive evaluation and comments. And we also agree with the limitation of the current mathematical model verification. We added this point in Discussion for a prospective future work (p.12).

REVIEWERS' COMMENTS:

Reviewer #1 (Remarks to the Author):

In this paper, using a novel approach, Seirin-Lee et al show that the pathological state of CSU can be inferred by geometric features of the skin eruptions. Their hierarchical mathematical modelling approach is unique and the analysis of 105 real CSU patients achieved 87.6% agreement by dermatologists, which may be useful for the clinical understanding of CSU.

Although the study is explanatory and should be confirmed using other independent data set, the authors' approach is novel, and they adequately responded to my specific comments.

I think the paper is acceptable for publication.

Reviewer #2 (Remarks to the Author):

The authors have sufficiently answered my questions.

Reviewer #3 (Remarks to the Author):

The review of the manuscript appears good and sufficiently descriptive. This work deserves publication.